# Intensification of Penaeid Shrimp Culture: An Applied Review of Advances in Production Systems, Nutrition and Breeding

**DOI:** 10.3390/ani12030236

**Published:** 2022-01-19

**Authors:** Maurício G. C. Emerenciano, Artur N. Rombenso, Felipe d. N. Vieira, Mateus A. Martins, Greg J. Coman, Ha H. Truong, Tansyn H. Noble, Cedric J. Simon

**Affiliations:** 1Livestock & Aquaculture Program, Bribie Island Research Centre, The Commonwealth Scientific and Industrial Research Organisation (CSIRO), Woorim 4507, Australia; artur.rombenso@csiro.au (A.N.R.); greg.coman@csiro.au (G.J.C.); ha.truong@csiro.au (H.H.T.); 2Marine Shrimp Laboratory, Federal University of Santa Catarina (UFSC), Florianópolis 88061-600, Brazil; felipe.vieira@ufsc.br (F.d.N.V.); m.aranha.martins@gmail.com (M.A.M.); 3Livestock & Aquaculture Program, CSIRO, Berrimah 0828, Australia; tansyn.noble@csiro.au; 4Livestock & Aquaculture Program, CSIRO, Queensland Bioscience Precinct, St. Lucia 4067, Australia; cedric.simon@csiro.au

**Keywords:** *Litopenaeus vannamei*, biofloc, RAS, BioRAS, microbial community, nutritional requirements, health, broodstock, additives, feed management

## Abstract

**Simple Summary:**

The shrimp sector has been one of the fastest-growing agri-food systems in the last 10 years. To overcome the increasing market demand, the transition to the intensification of shrimp farming is a reality in many countries. In addition, the desire to mitigate the risks posed by pathogens has driven many farmers to preference more controlled intensive systems with higher biosecurity. Shrimp nutrition and breeding are other areas that have directly enabled and improved intensification and will continue to be critical to ongoing growth in this sector. From this perspective, the aim of this review is to provide an update of the current production systems and strategies and explore the advances and key contributions that nutrition, breeding, and pathogen surveillance are having towards intensification and super-intensive shrimp culture.

**Abstract:**

Intensification of the shrimp sector, also referred to as vertical expansion, has been predominately driven by consecutive incidences of global disease outbreaks, which have caused enormous economic loss for the main producer countries. A growing segment of the shrimp farming industry has opted to use production systems with higher density, biosecurity, and operating control to mitigate the risks posed by disease. However, successful super-intensive shrimp production is reliant on an advanced understanding of many important biological and economic parameters in the farming system, coupled with effective monitoring, to maintain optimal production. Compared to traditional extensive or semi-intensive systems, super-intensive systems require higher inputs of feed, energy, labor, and supplements. These systems are highly sensitive to the interactions between these different inputs and require that the biological and economical parameters of farming are carefully balanced to ensure success. Advancing nutritional knowledge and tools to support consistent and efficient production of shrimp in these high-cost super-intensive systems is also necessary. Breeding programs developing breeding-lines selected for these challenging super-intensive environments are critical. Understanding synergies between the key areas of production systems, nutrition, and breeding are crucial for super-intensive farming as all three areas coalesce to influence the health of shrimp and commercial farming success. This article reviews current strategies and innovations being used for *Litopenaeus vannamei* in production systems, nutrition, and breeding, and discusses the synergies across these areas that can support the production of healthy and high-quality shrimp in super-intensive systems. Finally, we briefly discuss some key issues of social license pertinent to the super-intensive shrimp farming industry.

## 1. Introduction

The total global production of farmed marine shrimp increased 86% in the past 10 years, reaching more than 6.5 million tons in 2019 and a value of nearly 40 billion U.S. dollars. Countries in East and Southeast Asia (83.4% of production) and Latin America (16.3%) account for the major share of shrimp production, mainly based on two species—the Pacific white shrimp (*Litopenaeus vannamei*) with 83.1% of the production, and the tiger prawn (*Penaeus monodon*) with 11.8% [1]. To keep up with this growth, the industry needs to expand using sustainable production strategies [2]. Much of the industry growth over the past 30 years has been achieved through horizontal expansion, that is by expanding the footprint of low-input extensive and semi-intensive farming sectors [3]. However, vertical expansion, by means of increased intensification of farming, provides an alternative approach. Over the past 10 years, super-intensive farming of *L. vannamei* and high input practices have become more prevalent, which have been driven and enabled by the advancement of cutting-edge technologies and systems tailored to producing shrimp in high-density production systems [4,5].

Sustainable intensification is a promising approach to increase shrimp production, when there is increasing competition for the use of finite resources (e.g., land and water), but also when there is a need for a more ‘controlled biosecure’ environment similar to other intensive meat producers such as poultry and swine [6]. The success of sustainable intensification is dependent on the culture environment, but also on the biology of shrimp, with appropriate breeding and nutrition critical to supply quality animals and inputs into the super-intensive farming systems [7]. *L. vannamei* is the target species for intensification due to a range of favorable biological attributes, which include the lower requirement of dietary protein; anatomical features facilitating microbial particle grazing [8,9]; amenability to higher stocking density culture due to their gregarious nature; a broader tolerance to environmental parameters; and relative ease of domestication and thus selective breeding. In addition, the long history of breeding of the species has resulted in the development of a whole *L. vannamei* breeding industry, which consists of many competing breeding companies supplying quality and differentiated advanced breeding lines throughout the global industry [10]. All these factors, in combination with recent research efforts on intensification (Figure 1), explain the expansion of super-intensive farms using *L. vannamei* as compared to *P. monodon* and other penaeid species. 

The move towards increasing intensification by industry was not only fostered by increasing shrimp demand, but by consecutive disease outbreaks [11,12]. Multiple episodes of white spot syndrome virus (WSSV), acute hepatopancreatic necrosis disease (AHPND), enterocytozoon hepatopenaei (EHP), and white feces syndrome negatively impacted the main producer countries with substantial economic losses [13,14,15]. The desire to mitigate the risks posed by disease has driven many farmers to prefer more controlled intensive systems with higher biosecurity [16]. Moreover, these controlled environments can typically enable more crops to be harvested per year and optimize farm and land use [3]. As a result, super-intensive farms make up an increasing proportion of farms in high shrimp-producing regions (e.g., Southeast Asia; [7]). These super-intensive farms adopt two main approaches: (i) clear-water systems with high water exchange rates; and (ii) limited water exchange microbial-based systems. The first approach is easier to adopt and demands less technical knowledge, but relies on high inputs (e.g., energy for water renewal and circulation; and chemicals for water treatment) and poses higher disease risk. To overcome these issues of higher inputs and disease, the second approach is gaining more interest due to lower requirements for water exchanges, increased biosecurity, and providing more predictability and consistency [17].

Regardless of the approach adopted, one consistent characteristic of intensification is the increased inputs per unit area (e.g., per m^2^ or hectare). Compared to traditional extensive or semi-intensive systems (e.g., <30 shrimp m^−^^2^), higher levels of feeding, energy for water circulation and aeration, supplements (e.g., feed additives, water minerals, sanitation products, etc.), and labor are required in super-intensive systems. The increasing scale and complexity of inputs and operating parameters within the super-intensive systems exacerbate the importance of interactions among these inputs and parameters, and results in farming systems that are far more sensitive both biologically and economically, and which therefore require higher levels of proficiency in system management. Engle et al. [18] analyzed farms with similar management strategies in Thailand and Vietnam to demonstrate a linear relationship between intensification and inputs. Higher intensity/yield levels (i.e., low, medium, high, and very high) were associated with increasing levels of stocking density (shrimp m^−^^2^), feeding rate (kg/ha/crop), survival, aeration rate (hp/ha), and a greater number of crops per year. In most scenarios, the authors observed that economic outcomes improved with increasing intensification of production and resulted in greater yields (metric tons per hectare). These outcomes aligned with a fundamental realization that super-intensive systems require higher levels of system management expertise and effort than more traditional systems (Figure 2). For example, inadequate management of water quality affects the dynamics between microbial flocs and water parameters (e.g., nitrogen and oxygen levels) in the ponds, causing rapid deterioration of the environmental conditions. These inadequate water quality conditions negatively impact shrimp performance and health, causing immune depression and mortalities leading to significant economic losses. Therefore, the profitability of super-intensive farming can only be achieved through significant levels of technical expertise, and the employment of efficient and diligent management.

Shrimp nutrition and breeding are other areas that have directly enabled and improved intensification and will continue to be critical for the ongoing growth in this sector. In super-intensive conditions, feed quality, formulations, and management become more important as feed is the main source of nutrient input. Overfeeding can quickly overload the system and underfeeding or inadequate dietary formulations can result in nutrient deficiencies. In some cases, diets of low apparent digestibility and poor pellet integrity can compromise both growth and feed efficiency, as well negatively impact water quality, deteriorating the environmental conditions and animal health [19,20]. On the other hand, breeding has an important role to play to ensure that cultured shrimp supplied to super-intensive farms are well suited to the range of conditions likely to be experienced and that their production performance can be enhanced through an ongoing process of genetic improvement [21]. An understanding of synergies between the related areas of production systems, nutrition, and breeding is crucial to produce healthy, fast-growing shrimp, and to ensure the success of commercial operations. We connect these core areas in the context of ‘intensification’, where we specifically refer to super-intensive systems focused on *L. vannamei,* carried-out in fully lined ponds/tanks, and having culture conditions with high inputs (e.g., feeds, energy, supplements, and labor), high stocking densities (e.g., >150 shrimp m^−^^2^ during the grow-out phase), and a high level of technology applied. This article reviews the current production systems and strategies being used for *L. vannamei* super-intensive shrimp farming; explores the advances and key contributions that nutrition, breeding, and pathogen surveillance are having towards intensification, discusses the synergies across these different areas, and provides future perspectives for super-intensive shrimp culture.

## 2. Super-Intensive Production Systems and Strategies

Intensification of shrimp through super-intensive systems has been adapted for different regions, countries, as well as environmental conditions. Techniques vary depending on (i) the different ranges of salinity, latitudes, and temperatures, (ii) whether farming outdoors or indoors, (iii) where using single phase or multiple phase production, as well as (iv) the limitations of existing farm infrastructure, skilled personnel, and operational budgets and resources. In addition, the level of water exchange employed in super-intensive farming is highly dependent on location. In coastal regions, some level of water exchange and discharge is common, but an increasing number of farms are situated inland. These inland farms normally have the advantage of being close to markets, but typically rely on the reuse of water after multiple farming cycles due to water scarcity or restrictions [22]. Even in Europe, there is now a small level of commercial production in ‘boutique farms’ (i.e., 447 tonnes), with the main producer countries being the United Kingdom, Germany, and Austria [23]. Increasing competition for land and water in coastal regions will likely place more pressure for farms to be operated inland [24] and thus, the scale of inland shrimp production is likely to increase.

Despite these differences, many aspects of super-intensive farming remain constant. Production units (ponds or tanks) are normally fully lined (e.g., with high-density polyethylene HDPE), with different pumping and aeration lay-outs [3,25] to sustain stocking densities that vary from ~150 to more than 300 shrimp m^−2^ [26,27]. It is important to mention that intensification does not only refer to an increase in stocking density [18] but more broadly, to a sum of inputs that generate a higher biomass per unit area, expressed either as kg m^−2^, kg m^−3^ or tonnes per hectare. For instance, a survey of shrimp farmers in Vietnam found semi-intensive farms (considered a medium category in the study), with a single production phase and 90-day cropping cycle, stocked an average of 31 shrimp m^−2^ resulting in a final biomass of 0.35 kg of shrimp m^−2^ or 3.5 tonnes per ha per cycle [18]. With current intensification practices and increased stocking densities, the final biomass can easily surpass 3 kg of shrimp m^−2^. Krummenauer et al. [26] reported a final biomass of 4.1 kg of shrimp m^−2^ (41 tonnes per ha per cycle equivalent) using a 120-day biofloc single-phase culture. Indeed, these are only select examples, and yield achieved in any production system varies depending on a multitude of factors.

Many different systems and strategies are being used for super-intensive shrimp culture. In different farming regions, systems vary and include ‘pure’ (e.g., biofloc technology) to ‘hybrid’ (e.g., BioRAS) microbial-based systems, ‘mixed trophic’ systems, clear-water systems with high water exchange rates [28], and recirculating aquaculture systems (RAS) (more details in the Section 2.1 and Section 2.2). The operational strategies being employed for super-intensive farming vary from single-phase production to multiple phases production incorporating nursery systems [29] (detailed in Section 2.3). Regardless of the system or strategy, the intensification process has led to greater adoption of chemical sanitation protocols and pre-treatment of water, as well as heightened use of water supplements during the culture cycle. Water supplements are used to adjust and stabilize the water quality parameters, microbiological and environmental conditions, as well as to suppress the growth of pathogenic microorganisms [30]. Water supplements are being used for purposes of bioremediation (known as water probiotics; [31,32]), alkalinity adjustment [33] and to regulate mineral levels [34,35,36].

In the following subsections, we will explore the most common systems and strategies adopted (e.g., nursery systems) for super-intensive farming, including the use of integrated systems as a strategy to address the poor recovery of nutrients in many high-input shrimp culture systems.

### 2.1. Biofloc Technology (BFT) and Other Microbial-Based Intensive Systems

Biofloc technology (BFT) is most commonly used for shrimp culture [3,4,37,38] but is increasingly being used to culture fish and other aquatic species [29]. BFT differs from other production systems as it heavily relies on a beneficial and rich ecosystem of in situ microbes to minimize the need for water exchange [39,40]. While proficient microbial manipulation is the key to the successful operation of the system, appropriate engineering (e.g., pond layouts, aeration, pumping and drain systems), bespoke production management (e.g., water preparation, feed management, water quality monitoring and intervention, employment of biosecurity protocols), and having the necessary aquatic microbiological know-how (e.g., applying different strategies and water supplements aiming to develop the desirable microbial populations) are all essential for optimal functioning of these systems. Although a high level of expertise is required for all aspects of the system to work together, BFT systems are efficient production systems that can achieve resource optimization, in situ nutrient recycling, and generation of natural food sources through the formation of suspended microbial aggregates (i.e., bioflocs). These microbial aggregates, which may attach to hard surfaces or move freely in the water column [41,42], colonize the rearing environment and the gut of the shrimp [43], improving the activity of digestive enzymes [44], fostering shrimp and system health [45,46] and preventing disease outbreaks by in situ competition with pathogenic microbes or by reducing the virulence of such pathogens [46]. In this sense, the biofloc acts to suppress the pathogen load within the shrimp host and in the production environment [47].

The development of ideal ‘beneficial’ microbial communities is crucial in BFT [42], with bacteria generally considered the fundamental microbial group dictating system health. As per the ‘microbial loop concept’ [48], bacteria in BFT (especially the heterotrophic group) play a key role at the bottom of the food chain by utilizing dissolved organic matter. The bacteria are then consumed by protozoans, which in turn are consumed by larger organisms in the BFT food chain [49]. Bacteria can re-incorporate up to 50% of the carbon released by phytoplankton, accelerating mineralization and making the carbon available to higher trophic level organisms [48,50]. This recycling process is especially important in super-intensive conditions with limited water exchange and with high loads of nutrient inputs [51].

Recent research studies and reviews have detailed the microbial communities occurring in BFT and highlighted the importance of different strategies to develop, manipulate, and control the desirable microbial populations [3,4,9,29,52]. The major driver in BFT is the development of a microbial population dominated by heterotrophic and chemoautotrophic bacteria/organisms [53] and the control of algal blooms. A system dominated by photoautotrophic organisms such as algae is typically unstable, with large and sudden changes in water quality parameters, such as dissolved oxygen and pH, caused by rapid (and exponential) algal growth [42]. Beyond the rapid growth phase of the algae, the subsequent ‘crash’ of the algal population degrades the rearing system, as large quantities of dead algal cells and organic matter accumulate at the pond’s bottom, which promotes the spread of pathogenic bacteria such as *Vibrio* spp. [4].

From a nutritional point of view, a range of advantages of BFT have been reported and/or hypothesized including: (i) reduced feed conversion ratio [54,55]; (ii) reduced dietary protein requirement [43,56,57]; and (iii) a greater scope for alternative protein ingredients to replace conventional high-cost feed ingredients, such as fishmeal and fish oil [58,59]. Such advantages could be employed to decrease production costs in BFT culture and are enabled by the continuous availability and consumption of natural food sources in the form of the ‘bioflocs’ by the shrimp. Most publications on BFT systems have focused on *L. vannamei*, however, studies have also assessed the performance of other penaeid species in these systems [29]. Investigations have also reported that BFT promotes higher reproductive outcomes [60,61,62,63] and shrimp larval performance [64] as a result of better nutritional and sanitary conditions and enhanced immunity [65].

The BFT evolution over the past 20 years and related knowledge have provided a baseline for the development of other related microbial-based systems such as semi-biofloc [66,67], synbiotic systems [68,69], Aquamimicry [70] and AquaScience^®^ [71]. Nowadays, different microbial-based approaches can be found, all sharing key characteristics similar to BFT such as (i) pond systems with limited water exchange, high aeration and water movement; (ii) microbial aggregation and ‘flocs’ particle formation; and (iii) bacterial-based microbial manipulation: either by C:N ratio adjustments with carbon source application, and/or application of commercial microbial-based products (e.g., ‘water probiotics’). Figure 3 and Table 1 summarize the current microbial-based systems and their main characteristics, with many sharing the same characteristics as described previously for BFT [3,4,37,38]. However, some of the key differences operating in these alternative systems as compared to ‘pure/original’ BFT include: (i) reduced levels of suspended solids in culture water; (ii) reliance on chemoautotrophic bacteria and algae alongside heterotrophic bacteria to control toxic N-compounds; and (iii) utilization of fermented/pre-digested starch-based carbon sources (e.g., rice bran, wheat bran, and corn by-products) or vegetable nitrogen sources (e.g., soybean meal), with or without the addition of commercial products based on enzymes and blending of bacteria strains. Once added to the ponds, this ‘bacterial soup’ can promote zooplankton blooms (cladocerans, copepods, rotifers, insect larvae, among others), and help to control the algal blooms mainly due to the action of enzymes and other algicidal compounds excreted by bacteria [72], and increased turbidity. Finally, alternative systems can also be (iv) integrated with low trophic level fish species (more details in Section 2.4).

### 2.2. Water Exchange Systems: Flow-Through, RAS and Hybrid Systems

Flow-through or clear-water exchange systems have widely been used for commercial super-intensive shrimp culture (Figure 4). In large-scale operations, this type of system can be criticized for the large volumes of water required for production (i.e., commonly more than 5000 L water per kg of shrimp produced; [4]) and for the large amount of nutrient-rich effluent being discharged. In this regard, techniques with more efficient water use have been tested and developed and include recirculating aquaculture systems (RAS) and hybrid systems, e.g., BioRAS and green-water RAS. The RAS clear water-based technique is widely used in salmonid aquaculture and other high-value fish species [79]. However, the application in commercial-scale shrimp farming is relatively novel, especially in large operations [80], and there is limited information available (Figure 1). A laboratory-scale study with *L. vannamei* demonstrated positive results after 55 days of culture, in terms of survival (~78%), biomass produced (~2.0 kg m^−3^), and FCR (~1.5) [81]. In commercial super-intensive *L. vannamei* farms in Malaysia and Indonesia, large-scale RAS have been used to reduce the environmental impacts of wastewater discharge, but also to mitigate the risk of disease spread to other facilities caused by the high volumes of water released [80]. Instead of high-cost RAS equipment and filtering devices, these large operations normally recirculate water between different compartments of the farms, use settling basins and incorporate biological bioremediation via the culture of fish, mussels, oysters, and seaweeds, in these nutrient-rich waters for the purpose of suspended solids removal, biofiltration and nitrification [80].

In recent years, other practices have been developed combining the BFT and RAS in what is called a hybrid system. The hybrid system is a relatively new approach and there is currently limited information on the operation or efficacy of hybrid systems. The hybrid approach incorporates some RAS equipment and filtering devices (e.g., mechanical filtration, nitrification, denitrification, phosphate removal, ultraviolet UV and ozone systems) into BFT operations. Depending on the environmental conditions, such as levels of suspended solids, microbial management, C:N ratio, N:P ratio, and control of light intensity, the hybrid system can be photoautotrophic-based (green-water RAS) or heterotrophic-based (BioRAS). In both cases, the chemoautotrophic community (nitrifying bacteria) is expected to develop either in separated compartments (e.g., biofilters) or by attaching to the suspended particles in the water column [83]. An initial water preparation step (e.g., inoculating mature biofloc water or addition of specific algae and bacteria species) is commonly employed [84]. In a 70-day indoor pilot-scale study with *L. vannamei*, the BioRAS technique was tested using 30 m^3^ tanks coupled with a settling clarifier and nitrifying biofilter [83]. Shrimp were stocked at a density of 300 shrimp m^−3^ (~2 g juveniles), with biofloc-rich water from a shrimp nursery pond used to seed the nitrifying bacteria into the system and accelerate the start-up of the nitrification process. A growth rate of ~2.1 g week^−1^, a survival of ~93%, a yield of ~5 kg m^−3^, and an FCR of 1.6 were achieved. A commercial trial in Indonesia using BioRAS in a 110 m^3^ tank at a stocking density of 500 PL m^−3^ achieved a survival rate of 78% and biomass of 2.7 kg m^−3^ [84]. Different microbial constituents including nitrifying bacteria, the microalgae *Chaetoceros muelleri*, and the probiotic heterotrophic bacteria *Bacillus megaterium* were used for water preparation. In a 48-d *L. vannamei* nursery study, Tierney and Ray [85] compared hybrid (BioRAS), biofloc, and clear-water systems and obtained comparable results in terms of final weight (~0.6 g), survival (~80%), and FCR (~1.3). Table 2 summarizes some key characteristics of ‘clear-water’ flow-through, RAS, and hybrid systems, noting that there is limited literature on the operation and efficacy of these systems.

### 2.3. Nursery Systems

Intensification has not only been used for the grow-out phase of shrimp farming, but also for the nursery phase [88]. The shrimp nursery phase is a period of rearing between hatchery and grow-out, in which the post-larvae are maintained in special facilities for approximately 15 to 40 days, in one or two sub-phases (e.g., 14 + 28 days), and under high stocking densities (1500 up to 20,000 PL m^−3^) [4]. During this period, more precise management of feeding, water quality, pathogens, and larvae survival and conditions is required when compared to single-phase systems [89]. After approximately 15–40 days in the nursery phase, the juvenile shrimp are typically transferred to traditional earthen semi-intensive ponds or lined super-intensive ponds or tanks. The adoption of a super-intensive nursery phase can support more production cycles per year, optimizing the land use and improving the predictability and efficiency of production [4]. A study comparing the efficiencies of different *L. vannamei* farms in India found that enterprises using an intermediate rearing phase (i.e., a nursery phase) achieved a significantly higher production efficiency when compared to those that stocked post-larvae directly into grow-out ponds [90].

Table 3 presents recent studies (from 2017 onwards) evaluating different aspects of shrimp nursery systems focused on *L. vannamei.* These results indicated that the adoption of nursery systems can increase the efficiency of shrimp farming. Notably, high-density nursery systems using microbial approaches have also been successfully employed using *P. monodon* [91,92], and other penaeid species such as *Farfantepenaeus paulensis* [93], *L. setiferus* [94], and *F. brasiliensis* [95]. One study on *L. vannamei* BFT nursery systems reported no difference in PL performance when compared to a conventional microalgae-based system [96]. Tierney and Ray [85] compared BFT, RAS, and a hybrid nursery system for *L. vannamei* and found no differences in shrimp performance between the systems. Ferreira et al. [77] compared three different *L. vannamei* nursery protocols; a heterotrophic-based protocol, a chemoautotrophic-based protocol; and a protocol using mature biofloc water from a previous culture cycle. The authors found that the chemoautotrophic strategy reduced Vibrionaceae and resulted in improved shrimp growth when compared to the two other strategies. 

The use of artificial substrates in nursery systems has also been investigated. Greater yields were obtained by Legarda et al. [97] and Tierney et al. [98] when using substrates in BFT and RAS nursery systems, respectively. The type of substrate has been found to influence systems operation, as evidenced by Rezende et al. [99], who found that a polyester substrate (Needlona^®^) resulted in higher survival and lower concentrations of total suspended solids in a BFT nursery system compared to two other substrate materials (e.g., Bidim^®^ geotextile and mosquito net screen).

**Table 3 animals-12-00236-t003:** Summary of recent studies evaluating different aspects of *L. vannamei* nursery systems.

Production System	Evaluated Aspect	Main Findings	Reference
BFT	Different carbon sources	Lower ammonia concentrations in molasses and starch supplemented systems.	[100]
BFT	Stocking density and use of artificial substrates	Substrates increased shrimp yield.	[97]
BFT	Different artificial substrates	Needlona^®^ resulted in higher survival and lower concentrations of TSS.	[99]
BFT	Stocking densities	Optimum stocking density of 140 post-larvae L^−1^	[101]
BFT	Different BFT management strategies	Chemoautotrophic strategy reduced *Vibrionaceae* and improved shrimp performance.	[77]
BFT	Feeding frequency	Reducing feeding frequency did not affect shrimp performance.	[102]
BFT	Stocking density and water exchange	No water exchange did not affect shrimp growth.	[103]
BFT and microalgae-based system	Production system and TSS level	Both systems resulted in similar shrimp performances.	[96]
BFT, RAS, and hybrid system	Production system	No significant differences in shrimp performance between the 3 systems.	[85]
Hybrid RAS	Stocking density and use of artificial substrate	Higher shrimp yield when using substrates.	[98]

BFT: Biofloc technology; RAS: recirculating aquaculture system; TSS: total suspended solids.

### 2.4. Super Intensive Shrimp-Based Integrated Systems

A major challenge faced with the intensification of shrimp farming is the poor recovery of nutrients from feed, with only 23–31% of nitrogen and 10–13% of phosphorous typically recovered [104]. Therefore, most of the feed inputs going into the culture system deteriorate water quality, accumulating as organic and inorganic nutrient waste. When discharged, these nutrients can impact surrounding environments and natural water bodies [24]. Moreover, this nutrient discharge is a waste of costly inputs, and solutions are needed to alleviate the environmental impacts and increase shrimp production efficiency [105]. The co-culture of low trophic species combined with shrimp has the potential to consume a portion of the suspended or settled particles in the culture system, and act as a bioremediator against pathogenic organisms [106]. Moreover, there is a significant opportunity to co-culture plants with the shrimp, either using aquaponic techniques [107,108,109] or macroalgae integrated systems [110,111], with the plants serving to boost the assimilation of inorganic substances and so recycle the ‘waste’ nutrients. These approaches are known as integrated multitrophic aquaculture [105].

Recently, Poli et al. [112] examined the potential of fish to consume shrimp waste and enhance nutrient recycling. The authors evaluated tilapia (*Oreochromis niloticus*) reared with *L. vannamei* using biofloc technology and found that nitrogen and phosphorous recovery increased by 28% and 223%, respectively, when compared to a shrimp monoculture system. Holanda et al. [113] evaluated the potential of mullet (*Mugil liza*) to control the suspended particles in an integrated fish-shrimp BFT culture system. Shrimp performance was not impacted by co-rearing with the mullet, and lower total suspended solids concentrations were found in the system as compared to shrimp monoculture. Table 4 presents recent studies (from 2017 onwards) evaluating shrimp-based integrated culture. Although relatively new, this approach has shown promising results and more effort is needed to further develop these approaches as an avenue to improve the sustainability of super-intensive shrimp farming.

## 3. The Role of Nutrition in Shrimp Intensification

In super-intensive shrimp farming, feed quality (characterized by water stability, palatability, digestibility, and nutrient balance to meet the nutritional requirements of specific shrimp species), and feeding management (e.g., feeding ration and frequency to maximize feed availability and avoid deteriorating water quality) are crucial aspects to guarantee optimal culture conditions, growth, health status, and feeding efficiency. These are essential attributes to achieve profitability and sustainability in high-input systems as the compounded feed and supplements (e.g., feed additives) represent the vast majority of production costs for most intensive culture systems [18]. Typically, input costs (per hectare per year) have been shown to increase with the intensity of production. In these conditions, where the nutrients are mostly provided by the compounded feed rather than natural productivity [123,124], the key criteria to assess the cost of feeding should be overall production efficiency (e.g., cost per unit of retained protein and energy by the shrimp biomass generated), not just the cost of the feed. All these factors need to be considered in the context of shrimp intensification where the quality of compounded feed is more critical to the success of production compared to other traditional lower input systems.

The price of major raw materials used in shrimp aquafeed has been rising over the last 20 years (Figure 5). In this context, feed mills have focused on broadening their formulation portfolio to maintain feed quality and price. This increase in commodity pricing has not been reflected by the price of shrimp as a food commodity which has remained around USD 12 kg^−1^ over the last 20 years [125]. As a result, there has been a decrease in the shrimp to commodity price ratio indicative of tightening margins (Figure 5). The need to improve feed cost-effectiveness through ongoing nutrition and the use of alternative ingredients has never been greater.

Ingredients rich in protein, with balanced amino acid profiles, as well as marine sources particularly rich in essential micronutrients for shrimp, attract premium prices, so their judicious use in shrimp formulations is warranted to develop cost-effective feeds (Figure 6). Although considerable research advances have been achieved regarding shrimp nutritional requirements, feeding practices, and digestive physiology, further research addressing key nutrition knowledge gaps is needed to ensure the sustainability and efficiency of intensive shrimp farming [126].

It is unknown whether nutrient demands of shrimp are met under super-intensive conditions, though it is likely that the provision of essential macro/micronutrients and modified feeding management strategies could assist animals to cope and thrive in the more challenging super-intensive farming environment. However, in these conditions, additional caution is required due to higher concentration of excreted nitrogen resulting from protein catabolism, increasing the potential of environmental impacts. There is an ongoing need for alternative feeding strategies and feeds to be tailored to a particular production system. This section covers aspects of shrimp nutrition particularly relevant to super-intensive systems compiled into a series of topics addressing key nutrition challenges and novel applications namely: nutrient requirements in super-intensive systems and tailored feeds, digestible ingredients and pellet stability, marine and microbial-based growth promoters, feed additives, and feed management in super-intensive systems.

### 3.1. Nutrient Requirements in Super-Intensive Systems and Tailored Feeds

The nutrient requirements of *L. vannamei* have only been assessed in a few commercially relevant rearing systems differing in stocking density, salinity, and other water conditions [126,127,128,129,130]. The effects that different rearing conditions have on dietary requirements for many nutrients known to be required by *L. vannamei*, including fatty acids, some amino acids, vitamins, and minerals, have not been reported. Furthermore, the environment in which these more ‘commercially relevant’ nutrient requirement values are obtained, is not consistent. The effect of intensification on nutrient requirements has not been assessed directly, but it is possible that the presence of increased stressors from higher stocking densities and reduced availability of natural feed, in systems such as RAS, will result in higher demands for certain micronutrients. Indeed, requirement values available from IAFFD show that the requirements of shrimp in different production systems will differ subtly (Table 5). *L. vannamei* reared in semi-intensive systems will have lower requirements of protein, energy, and lipids compared to those reared in RAS and intensive systems. Published literature showed that *L. vannamei* reared in semi-intensive systems achieve their highest weight gain with 32.9% dietary protein. In the same study, shrimp reared in biofloc grew well when diets contained 30.3% dietary protein [57], suggesting lower protein diets can be used in systems with higher availability of natural feeds such as biofloc. Higher dietary lipid (relative to higher lipid to protein ratios) has been shown to improve shrimp resistance to oxidative stress and immune system pressure [131]. Requirement studies may need to be specifically designed to assess the need for micronutrients within those conditions (likely lower levels of individual feed intake). This was demonstrated in a recent study where methionine inclusion levels to achieve maximum growth increased with stocking densities over the range of 50 to 100 shrimp m^−2^ [128]. An interaction exists between stocking density and methionine content driven by natural food availability, and this will likely impact the need for higher protein content when amino acid profiles are not supplemented with crystalline amino acids. More research is needed to better understand the nutrient requirements of shrimp across the different intensive farming systems. Table 5 shows an example of different nutritional requirements for *L. vannamei* according to different production systems.

Many aquafeed companies recognize the need to support the development of the super-intensive farming industry by developing specific feeds for different high-inputs systems [133,134]. However, this is a difficult task due to the many management practices that will affect the nutritional strategy in shrimp feeding. For example, the use of BFT in super-intensive systems has a critical implication on the efficacy of the formulated feed. Biofloc is considered a good source of amino acids, fatty acids, vitamins, and minerals, and contains polymers, exopolysaccharides, organic acids, and immunostimulatory compounds that benefit shrimp’s immune system [135]. Therefore, the nutrients provided through the microbial aggregates (bioflocs) suggest there is potential for sparing some of these nutrients in diet formulations [136]. If not pared back, the excess of nutrients from the combination of formulated feed and bioflocs can lead to elevated levels of nitrogen and phosphorous in the culture system [9]. Thus, diet formulations and feeding regimes for BFT systems have been developed to account for biofloc composition and its variability, and the difference in formulated feed consumption between these systems. Pellet feeding rations in BFT systems can be reduced by 15–30% [137] due to the nutritional contribution of floc consumption [54]. The nutritional supplementation of BFT can allow a reduction in the protein content of fed diets [41,136,138] and allow higher inclusion levels of alternative ingredients [139,140]. While ‘standard formulations’ designed for traditional systems might be convenient in terms of feed mill logistics, it is likely that tailored feeds can lead to improvements in feeding efficiency and water quality in the grow-out environment. A trend is already emerging where there is an increasing number of bespoke feeds being developed to accommodate for unique super-intensive production systems [133]. However, the clear caveat is that the efficacy of these feeds requires close consideration of the system they were developed for. Until we are able to fully understand the influence of management practices and feed formulations on shrimp and the rearing system, it is unlikely that there will be a single optimal feed formulation for all super-intensive farming systems.

### 3.2. Digestible Ingredients and Pellet Stability

Pressure to reduce the reliance on fishmeal in shrimp feed has increased the diversity of alternative protein sources being considered for aquafeed production [141]. However, typically these alternative protein sources do not possess an ideal amino acid profile, do not have the same desirable digestibility, palatability, and contain more anti-nutritional factors than fishmeal [142]. Such anti-nutritional factors include enzyme-inhibitors, unpalatable compounds, nutrient-binders, and gut irritants [143]. There is growing evidence that some shrimp species can be more susceptible to anti-nutritional factors than others. For example, high plant-based diets were shown to impair the growth and utilization of protein and energy in *P. monodon* [144]. However, some studies with *L. vannamei* have revealed successful replacement of fishmeal with no adverse effect on growth, survival, and gut health [145,146,147,148].

The apparent digestibility of various feedstuffs has been determined for *L. vannamei*, including those of plant and animal origin (Table 6). Some examples include meat and bone meal [149,150], feather meal [149,151], poultry meal [149,150,151,152,153], soybean meal [150,151,152,154], soy protein isolate [149,151,154], canola meal [150], and wheat flour [155]. Most of these studies compared the digestibility of these feedstuffs with fishmeal. In addition, the use of synthetic amino acids in shrimp has shown promise as a strategy to balance the amino acids profile of diets containing alternative protein sources and improve their utilization [156]. *L. vannamei* fed low fishmeal-based diets supplemented with synthetic amino acids achieved similar growth performance as those fed the high-fish-meal control [157]. Thus, appropriate supplementation of synthetic amino acids can be useful in diets using alternative protein meals as shrimp have a requirement for a balanced amino acid profile, rather than for crude protein. However, regardless of the source of alternative proteins, the differences in nutrient digestion need to be carefully evaluated and are important considerations when selecting ingredients for super-intensive shrimp aquaculture. In some cases, alternative sources that are poorly digestible can compromise both growth and food conversion efficiency [158,159,160], as well as negatively impact the water quality (e.g., high levels of suspended solids, toxic nitrogen compounds, and phosphorous load) [9]. Protein retention efficiency of shrimp is low, ~10–25% [161] with fish achieving two to three times this. The replacement of fishmeal with plant-based ingredients often worsens protein retention efficiency [162,163] and the consequential metabolic cost and increased effluent need to be managed, especially in an intensive system.

The same principle can be extrapolated for diet stability. High stability of diets in water becomes increasingly important in super-intensive systems where the breakdown of uneaten feed particles can lead to major issues with water quality [165]. A more stable diet matrix can be achieved by functional ingredients like diet binders, and starches which gelatinize and entrap water-soluble ingredients. Diet binders available for shrimp feeding include synthetic chemicals (e.g., bentonites, hemicellulose, carboxymethycellulose, and urea-formaldehyde mixtures) and natural extracts (e.g., algae hydrocolloids such as agar, carrageenan and alginate, terrestrial plant pectins, glutens, and starches). Such pellet binders were shown to have varying effects on improving water stability and reducing nutrient leaching [166]. On the other hand, some binders were observed to have a negative effect on crustacean growth [167]. Thus, effective binders for shrimp feeding need to achieve the right balance between water stability and nutrient bioavailability. It is also important to assess the interaction of the carbohydrate source on the usefulness of pellet binders and the overall outcome on pellet water stability, texture, and nutrient availability. Selection of carbohydrate ingredients with high digestibility and resulting pellet stability would be a useful strategy to manage diet quality for super-intensive systems shrimp diets.

### 3.3. Marine and Microbial-Based Growth Promoters

Optimal growth is crucial in super-intensive systems, as it can directly impact the number of potential culture cycles per year, optimize the farm resources and improve profitability. In this context, one option would be to include growth promoters in feed formulations. ‘Unknown growth factors’ from specific ingredients in shrimp diets, such as some marine invertebrate-derived meals and hydrolysates including squid, krill, other crustaceans, and also some from microbial origins, have been the subject of research in the last 30 years [135,168,169,170,171,172,173,174,175,176,177,178,179,180,181]. Premium commercial shrimp diets often rely on a selection of these ingredients to maximize attractiveness, palatability, and growth performance. This is particularly the case for *P. monodon*, which generally requires greater levels of these ingredients to sustain optimum growth, survival, and health, as compared to *L. vannamei*, and particularly when fed low fishmeal diets [161]. The actual biological mechanisms by which these ‘growth factors’ promote culture performance remain poorly characterized. For krill meal, a growth factor was isolated from its insoluble protein fraction [177], and krill meal was demonstrated to have a positive effect on feed palatability through increasing the feeding duration [182]. For marine microbial biomass, a range of potential bioactive nutrients has been measured [135]. Recent evidence indicates that the commercial microbial biomass Novacq^TM^, as well as squid meal, have several modes of action, acting on feed intake rate, gastro-intestinal transit rate, and amino acid absorption rate; and which collectively increased the retention efficiency of protein and lipid for growth [173,183]. Inclusions of up to 10% Novacq^TM^ increased growth and feed conversion efficiency at a range of fishmeal and protein inclusion levels in *P. monodon* and *L. vannamei* by as much as 60% against a control diet in clear-water systems [173,180,184,185]. The judicious use of additives with growth stimulating properties is relevant for intensification, where stocking densities are too high to rely on supplementation from natural food sources. They also allow better performance of aquafeeds with limited to no fishmeal [180,186]. Higher feed costs are justified in efficient intensive production systems when the following outcomes can be achieved: improved culture performance enabling improved survival and yield, shorter crop cycles allowing for more subsequent crop cycles within the year and/or increases in the final size class of the product to reach premium prices. Future economic modeling to decide on the target market segment with the best economic return for set feed prices is key.

### 3.4. Feed Additives for Improved Nutrition and Health

In super-intensive systems, shrimp are continuously challenged due to high stocking densities, high feeding volumes, and high-water nutrient loads which can collectively act to unbalance the water microbial profile (causing e.g., Vibrio outbreaks), affecting growth, health, and suppressing the shrimp immune status. To minimize these negative impacts, a routine strategy has been the application of (i) water supplements (e.g., water probiotics and sanitation products) but also (ii) diverse feed additives either added during pellet processing in feed mills or top-coated on farms. This last procedure is typically carried out by mixing the additives with water or ‘binders’ (e.g., fish oil, sugar-cane molasses; or starch-based commercial products), then top-coating onto the commercial feeds. This process could have some negative impacts through (i) altered nutritional composition of the feeds (ii) reduced feed consumption; (iii) reduced pellet integrity/stability; and (iv) uncertainties as to whether the desired dosage has been properly delivered.

According to Mordor Intelligence [187], the shrimp feed additives market was valued at USD 66 million in 2018 and estimated to reach USD 104 million by 2024, registering a CAGR of 7.8% during the forecast period 2019–2024. This increasing trend is likely supported by the super-intensive operations that strongly rely on feed additives, stimulating shrimp immunity, and maintaining a healthy shrimp and pond environment. The pond water bacterioplankton composition has been suggested as a potential indicator of shrimp health [188], and a suitable profile has been shown as an effective tool to avoid disease outbreaks. Several feed additives and immune stimulants are now available [189], although there is limited data on their effectiveness under commercial conditions, especially in dynamic (microbial) super intensive conditions.

There are a large variety of feed additives available to shrimp farming, including minerals, vitamins, fatty acids, amino acids, organic acids, phytobiotics, essential oils, specific microbial and marine polysaccharides, nucleotides, pigments, prebiotics, and probiotics [181,190,191,192,193,194]. In the context of super-intensive systems, a recent review [195] revealed feed additives have an important role in minimizing antibiotic use in shrimp culture and other aquaculture industries as well as reducing the incidence of disease and promoting growth. The mechanisms by which feed additives have such beneficial effects on shrimp health include stimulating the innate immune system, providing micro/essential nutrients, and maintaining a healthy microbiome. For example, sulphated polysaccharides derived from seaweed (e.g., *Gracilaria* sp.) and phloroglucinol added to shrimp feeds stimulated several innate immune parameters and led to greater resistance to bacterial and viral agents [196,197,198,199]. Lipopolysaccharides coated on feed pellets were shown to increase shrimp survival following exposure to *Vibrio harveyi* [200]. Organic acids or short-chain fatty acids are one of the most commercially used additives in shrimp feeds due to their relatively low price and known positive effects on growth and survival, gut integrity, the immune system, and disease resistance [201,202,203,204]. Feeds containing organic acids are commercially used and marketed to assist shrimp to tolerate *Vibrio* spp.; a ubiquitous microbe that causes major loss by gradual, but consistent mortality. Many shrimp diseases are commonly associated with a dysbiosis of the gut microbial community [205] and the ability to regain a normal community may enable a shrimp to overcome or tolerate pathogen infection. Thus, feed additives may also promote a healthier more resilient microbial community [206,207]. For example, *P. monodon* fed the diets containing the additive Novacq™ for 6 weeks resulted in several *Vibrio* operational taxonomic units (OTU) with lower relative abundance compared to the non-additive control [173]. The efficacy of feed additives on shrimp health varies with species, production system, location, feed additive type, inclusion level, feeding duration, interaction with other additives, and mode of incorporation into the feed [208].

Table 7 illustrates some examples of commercial feed additives available in the shrimp market. We invite readers to discuss these products with commercial suppliers including published and unpublished efficacy studies. The examples listed are not meant to exhaustively cover all products available, and the authors are unaware of a suitable published review focused on their effectiveness in shrimp. In some cases, application of the products is not exclusive to shrimp farming, and insufficient knowledge of the mechanisms in which they act on shrimp, as well as the uncertainty associated with administration (mode, doses, and frequency) under different culture phases and pond/environmental conditions, likely constrain effective adoption of feed additives [208]. This represents a challenge for the farmers and the broader industry. The functionality of feed additives across the diversity of production systems and environmental conditions needs to be further explored and understood. Moreover, further research focused on tailoring the types of additives appropriate for different super-intensive systems, identifying appropriate inclusion levels of these additives, and improving our understanding of the interaction among the different additives, is warranted.

### 3.5. Feed Management in Super-Intensive Systems

As formulated feed is a major production cost of shrimp farming, the quantity of feed and frequency of feeding are key factors that drive economic success or loss in any production cycle [18]. Several studies have demonstrated improvement in growth and feed efficiency when shrimp are fed multiple times a day [184,209,210]. Improvements in growth performance without a detrimental effect on the shrimp nutrient retention efficiency were achieved in research trials using premium feeds fed *ad libitum* [173]. The digestive physiology of shrimp is well suited to continuous feeding, due to the rapid rate of foregut filling (within 30 min) [161] and intestine evacuation (i.e., 22% per hour in large juveniles) [183,211]. 

Most shrimp farms still rely on human labor to gauge feeding rates and to feed shrimp, which restricts feed delivery frequency. The use of novel feeding technology, such as timer-feeders and acoustic demand feeders, can dramatically improve growth and feeding efficiency while reducing labor costs [212,213]. Careful management of the amount and timing of feed ration delivery is particularly important in the context of low water usage super-intensive systems where the lower rates of water exchange rates mean that overfeeding results in the rapid accumulation of organic matter, dissolved nutrients and can lead to pathogens outbreaks. Refer to Darodes de Tailly et al. [214] for a recent review of advances in our understanding of shrimp feeding behavior and feeding methods.

Another avenue to maximize nutrient bioavailability is to formulate feeds that are more stable post-immersion in order to deliver a suitable nutrient balance over a longer period of time. While shrimp have been found to uptake and assimilate crystalline amino acids well, these amino acids can leach out of diets within minutes and any lag in the feeding response will reduce the assimilated concentrations [161,215]. A cost-effective commercial technology to bind low molecular weight water-soluble nutrients is currently not available but warrants ongoing research. Additionally, in traditional pond-based systems, feeding restriction can lead to improved consumption of available natural food sources and lowering of the FCR, especially early in the culture cycle [216]. Previous clear-water studies in tanks have also found that feed efficiency improves with restricted feeding (50–80%) in *L. vannamei* [217] and in *P. monodon* [173,184,218] due to higher digestive enzyme activities and a change in the gut microbiota. This suggests a higher capacity for digesting nutrients when fed restrictively. However, in all the above studies, feed restriction reduced growth performance, and this effect would likely be exacerbated in intensive culture systems where animal access to feed can be limited [54]. While efficiency can be improved through restriction, a full cost–benefit analysis needs to be performed to understand the overall production cost of such an approach, especially in super-intensive conditions with high daily operational costs.

## 4. Breeding and Pathogen Surveillance in Shrimp Intensification

Breeding has an important role in super-intensive farming to ensure that the cultured shrimp stocked into the rearing systems are suited to life in a more crowded and nutrient and microbial laden environment [4], and to ensure that the shrimp performance can be enhanced through an ongoing process of genetic improvement. Throughout this section, we refer to ‘breeding’ in its broadest context and cover the topics of domestication, health and health status, biosecurity, and genetic improvement, and we relate these topics to their importance for super-intensive shrimp culture. We limit our discussions to *L. vannamei*, a species for which intensive farming is more common, and commercial breeding more advanced.

### 4.1. Domestication and SPF

The need for clean pathogen-free shrimp is critical for super-intensive farms. The crowding and higher nutrient and microbial loads occurring in these super-intensive systems typically impose increased stress on the animals and an environment conducive to pathogens, which if present in the animals, can replicate and manifest into disease [219]. Stocking of pathogen-free shrimp along with other effective management strategies, therefore, reduces the risk of disease. Pathogen-free shrimp, appropriately referred to as specific pathogen-free (SPF) shrimp, can only be developed through domestication; domestication is the process where an animal’s life cycle is closed within a captive environment. For a more in-depth review of the SPF process and terminology see [6]. *L. vannamei* domestication and breeding programs commenced in the late 1980s with commercial domestication achieved by the early 1990s [6,220]. The initial driver for domestication was the industry’s need for commercial broodstock free of certain pathogens commonly found in the wild broodstock [221]. Disease risk was mitigated, if not wholly removed, by breeding companies developing SPF domesticated breeding lines maintained in highly biosecure facilities referred to as Nucleus Breeding Centres (NBC) which supplied the “clean” broodstock (or putative broodstock) to the hatcheries supplying the postlarvae being grown in commercial ponds [222,223]. Importantly, while SPF lines provide an ideal starting health status for farming, these stocks are still vulnerable to pathogen infection once in the ponds.

As shrimp lack adaptive immunity and rely on innate immune responses [224], traditional vaccination is not possible to manage disease [225,226]. Therefore, for the pathogens that posed the greatest disease risk to shrimp farming, exclusion through the use of SPF breeding lines has provided the central pillar to mitigate disease risk. Notably, much research since has focused on developing means to reduce, and ideally clear, pathogens from infected shrimp using RNA interference (RNAi) [227,228,229,230,231,232,233,234,235], however, there are no published accounts of such methods being applied routinely to achieve SPF status in commercial breeding lines. Recent research suggests there may be other therapeutics and novel strategies that offer the promise of improved disease outcomes for breeding lines such as selection for the presence of protective EVEs (endogenous viral elements) and “New circular DNA vaccines” [236] or transgenerational immune priming (see review by Roy et al. [237]). Yet despite these recent research findings, the SPF concept currently continues to be a dominant strategy for disease management, typically coupled with other means that provide disease mitigation, such as on-farm biosecurity, improved nutrition, best management rearing practices to reduce stress on animals, and genetic selection for pathogen tolerance or resistance (discussed further on) [16,221,238].

While domestication provided the vehicle to achieving SPF, it has also opened opportunities for other breeding advances through genetic change within the cultured shrimp breeding lines. Genetic improvement has been achieved through ‘directed genetic selection’ for traits yielding production and financial benefits within commercial farming environments. ‘Domestication selection’ is another process of genetic change, a process by which animals adapt to captive life [239], which has been fundamental to achieving domestication and the overall genetic improvement of shrimp. Changes in the genetic characteristics of animals arise through selection for “inadvertent” or “ancillary” traits which occur as the animals are being domesticated, but which are very beneficial to a future captive life, such as heightened docility in broodstock systems and the ability to reproduce in captivity [240,241]. Such changes result from the inherent selection pressures applied in achieving ‘closed-life cycle’ rearing; whereby an animal must be able to survive and then reproduce to be domesticated. Importantly, such domestication selection has both enabled successful domestication of *L. vannamei* but has also been critical in enabling success with ‘directed genetic improvement’ that has underpinned shrimp breeding progress, and together with the development of the SPF processes, contributed so significantly to the enabling of super-intensive shrimp farming.

### 4.2. Biosecurity and Pathogen Surveillance

Beyond the SPF process whereby pathogens posing risk are excluded from the breeding population being maintained within NBC’s [223], other aspects of health and biosecurity are critical to ensuring ‘pathogen freedom’ is maintained moving through the subsequent tiers of the production chain; through the Broodstock Maturation Centres (BMC’s), the commercial hatcheries and then the commercial grow-out farms. However, it is increasingly challenging to maintain pathogen freedom moving through these production tiers. As discussed in earlier sections, some super-intensive farming systems such as biofloc, RAS, and hybrid systems, can provide added biosecurity measures that minimize incursions of pathogens from external sources. However, even in these systems more omnipresent microbial species (e.g., *Vibrio* spp.) are difficult to keep out and can at times compromise the health of the shrimp [30]. Consequently, there is an increasing need for pathogen surveillance and further disease mitigation strategies that improve the overall health and immunity of shrimp in super-intensive systems.

Pathogen surveillance is a key element of farm biosecurity and is even more critical in super-intensive systems where rearing stresses and economic risks are heightened. Early and accurate detection of pathogens may allow for some intervention strategy to be applied (e.g., adjusting doses of water probiotics or feed additives) and/or containment of further spread. There are several approaches for pathogen surveillance including point-of-care (POC) methods that can be used by farmers (if jurisdictional laws permit) to obtain real-time data [242] and private and government laboratory services based on PCR or histological technologies. Each of these methods has its benefits and challenges, for example, many of the POC methods use antibody-based technology that may not be as sensitive as PCR-based methods [243]. However, technological advances have brought about several portable devices equipped with PCR and real-time PCR capability meaning highly sensitive and even quantitative testing of samples can be carried out pond-side by farmers. High-throughput sequencing (HTS) is also becoming more accessible and applied to aquaculture systems. The use of HTS can provide insights into the microbial community composition and function within the system itself, and within the cultured animals, and provide further context around disease manifestation commonly referred to as the “pathobiome” [244]. These new technologies present a significant opportunity to improve pathogen surveillance in super-intensive farming systems. Certainly, further development and application of these new technologies to improve our understanding of these complex super-intensive microbial systems will be important to determine how they can be manipulated for better health and productivity outcomes.

### 4.3. Genetic Improvement

In addition to the important role that breeding has in supplying shrimp free of pathogens that pose risk in grow-out, breeding allows stocks to be improved for genetic traits that are desirable for farming [223]. For super-intensive farming, such ‘genetic improvement’ enables the development of stocks that grow fast and survive well in the more crowded, and nutrient and microbial-rich intensive farming environments. Given the higher production and investment risk posed by pathogens as farming intensities increase, breeding has allowed stocks to be produced with reduced susceptibility to pathogen infection and severity of disease when reared in these super-intensive systems. There are now many commercial *L. vannamei* breeding companies focusing genetic improvement on SPF lines bred for faster growth and disease tolerance traits [10]. While the economic importance of genetic improvement for growth is evident across the shrimp industry, and the basic approach to genetic improvement for growth is well-established and relatively inexpensive for breeding companies to make improvements [225], breeding for disease tolerance traits is more costly and complex [245], and can require breeders to consider multiple pathogen candidates.

Genetic improvements for specific pathogen tolerance traits in *L. vannamei* have now been achieved for a range of pathogens, despite the ‘heritability’ estimates for many pathogens being low to moderate [16]; the heritability being a key breeding statistic quantifying the degree of genetic control over the observed phenotypic measure of a trait, such as survival post-challenge. The first major success with genetic improvement for disease traits in shrimp was realized through the development of *L. vannamei* lines tolerant to Taura Syndrome Virus (TSV) two decades ago [223,246,247]; this success ultimately eliminated TSV as a major threat to the industry. However, for other shrimp pathogens, such as White Spot Syndrome Virus (WSSV), tolerance has proven harder to achieve and taken much longer to develop in commercial breeding lines. For WSSV, the very high mortality rates experienced in challenges made it difficult to ‘discriminate’ survival among families, which contributed to very low estimates of heritability and made it harder to make genetic progress [248,249,250,251,252]. Notably, both the promise and success of WSSV tolerance have now been demonstrated through research lines and publications (e.g., G. Lo, personal communication in [6,226,253]) and realized through WSSV-tolerant lines now entering the market [10]. Commercial breeding lines are now being marketed as having tolerance for a range of other pathogens including infectious hypodermal and hematopoietic necrosis virus—IHHNV, and acute hepatopancreatic necrosis disease/early mortality syndrome—AHPND/EMS, while still more companies are marketing lines with generalized disease tolerance and resilience [10].

Breeding for both growth and disease tolerance, in theory, provides a way that shrimp breeders can supply lines ideal for any farming environment. However, to develop lines that perform well for different traits (e.g., growth and pathogen tolerance), across different environments, and for different production systems and intensities, breeders need to know how these traits (and traits measured across different environments) are associated or ‘genetically correlated’ to each other. Traits may be positively associated, with selection for one trait resulting in indirect improvement in the other, with one such example being improved survival and harvest weight under conventional commercial grow-out conditions [254]. Traits may have no correlation, thus selecting for one has no impact on the other trait and both may be improved in parallel. Given the range of systems and geographies over which shrimp farming occurs, breeders must consider whether they can develop lines that perform well across the breadth of farming environments, or whether specialist lines are required for different traits and environments. For breeding lines focused on growth, it is not uncommon that enhanced performance is claimed across a variety of production systems; though certainly some breeding lines are marketed as being selected for specific environments, including super-intensive farming systems [10].

Besides pricing and availability constraints, the choice of breeding lines that super-intensive farmers might decide on will be influenced by the risk posed by pathogens and other stressors within their farming system. This depends on the level of biosecurity and management able to be maintained in their farming system and their appetite for risk, as well as the trade-off in reduced growth if using disease or stress-tolerant lines [16,245]. Different *L. vannamei* breeding companies are now offering a range of breeding lines selected for either or both growth and tolerance to disease [10]. Commonly companies offer separate lines focused on either growth or pathogen tolerance traits; however, there are some lines selected for both growth and a single pathogen tolerance, and some for growth and multiple pathogen tolerance. Recently, some companies have developed breeding lines focused on general resilience, rather than specific pathogen tolerance. Companies now offer lines specialized for (i) different environmental conditions, such as tolerance to lower water salinity or performance at higher densities; or (ii) growth on diets using lower levels of animal protein inputs [10]. Due to the higher rearing densities and financial investments per crop, most super-intensive farms will use SPF breeding lines to mitigate disease risk, with lines both SPF and genetically tolerant to pathogens used to provide further ‘genetic insurance’ against disease.

Given the expected growth of the super-intensive grow-out industry sector, combined with the recent shift towards breeding companies producing a greater diversity of lines to suit different farming systems, it is likely that super-intensive shrimp farmers will in the future have an even wider range of breeding lines to choose from. In some instances, for farming operations where the highest levels of biosecurity can be maintained at very high stocking densities (e.g., >300 shrimp m^−2^), breeding efforts may concentrate on faster-growth as a means to ensure profitability in these very high investment systems [10]. However, breeding lines with enhanced tolerance and resilience traits will likely also have an important role where farming operations are less amenable to high-level biosecurity throughout their crop, as insurance to mitigate pathogen incursions and disease.

## 5. Sustainability and Social License

As for other aquaculture industries, the super-intensive shrimp farming industry needs to consider the many dimensions of sustainability and social license that both provide opportunity, but also potential barriers, to the development of the sector, and particularly the acceptance of the shrimp product by global markets. In this brief section, we note select examples of the positive attributes that super-intensive farming presents in terms of environmental sustainability, and of challenges facing the sector from an animal welfare perspective, for the purpose of highlighting aspects of the industry that have the potential to either bolster or hinder future development.

From an environmental sustainability perspective, current super-intensive systems (i) use less water per kilogram of shrimp produced [4], (ii) reuse water [22], (iii) have lower FCR [54], and (iv) optimize farmland and water resources [3]. Consequently, for efficient commercial operations, there are many environmental benefits of using super-intensive shrimp farming approaches on grounds of resource efficiency. Of course, critiques of the sector can be made based on the ongoing use of ‘wild-caught’ fishery products within many of the diets used in these super-intensive systems. However, a counter perspective is that many such systems (e.g., BFT) allow more environmentally friendly diets and feeding regimes to be employed. Moreover, there is much scope and interest to improve the environmental quality of diets used in the super-intensive farming systems.

From an animal welfare perspective, research has shown that at high stocking densities (e.g., increasing from 300 to 400 and 500 shrimp m^−2^) the activities of digestive enzymes (amylase, trypsin, and lipase) are reduced within the culture shrimp, as well as the immune status, leading to mortalities when challenged with *Vibrio harveyi* [255]. Further research to understand the impacts that high-density rearing has on shrimp is warranted to develop rearing methodologies that foster improvements in immunological and health outcomes of the shrimp.

Looking more broadly across the shrimp industry production chain, the shrimp hatchery sector that supplies the super-intensive farming industry is commonly criticized on animal welfare grounds due to the common use of unilateral eyestalk ablation of female broodstock; this technique used for the purpose of stimulating female gonadal development and synchronizing egg and nauplii production in hatcheries. ‘Ablation’ is of growing consumer concern and currently prohibits acceptance of products into certain markets, and moreover, there is growing pressure on producers and retailers to provide products from supply chains that do not use this practice [256]. While progress towards eliminating the need for the technique has been made by companies using alternative ‘natural approaches’ to breeding [257], these companies currently supply a small fraction of the global market and focus mostly on niche organic markets. Other alternative approaches that could be deployed by hatchery-only operators, and so used more widely throughout industry, such as RNA interference injection methods [258,259,260,261], have shown some research promise, but the authors are not aware of any commercial use of such methods. While the welfare concerns over ablation are not unique to the super-intensive farming sector; being of wider relevance for much of the global shrimp farming industry, there would still be a significant benefit in the super-intensive sector working with breeding companies to develop alternative approaches to de-risk future and growing consumer concerns over the practice, but also as an undertaking to modernize the shrimp industry and build-on some of the positive social license credentials of super-intensive farming.

## 6. Conclusions and Future Perspectives

Intensification of shrimp farming is a reality, and a step back from this industry trajectory is unlikely. Regardless of the production system chosen and adapted, it is crucial to tailor the management strategies and practices adapted to local conditions (e.g., environmental, technological, financial, and human resource capability). Proper interventions supported by holistic monitoring and interpretation of the different water quality, production, and environmental parameters will contribute to yield consistency and reliability; key drivers of success in commercial super-intensive operations. Considering the high load of nutrients in the form of uneaten feed, feces, and diverse organic matter in super-intensive systems, from an environmental and social license perspective, there is a need to convert these outputs into high-value products such as microbial biomass or complementary aquatic protein, by applying circular economy approaches. Research efforts to develop multitrophic aquaculture systems, which integrate shrimp with other aquatic organisms and plants, need greater focus. In addition, with the progress of genetic and other ‘omic’ tools, more refined microbial assessment, functional understanding, and ability to modulate communities will be increasingly feasible and provide more predictability and consistency of the culture conditions. In the same context, precision farming with automation, new sensor technologies, and decision support tools [17] will provide efficient management of the commercial ponds for healthy production of shrimp; the key mantra of success in this industry.

Effective intensification of shrimp farming, as in other cultured terrestrial and aquatic species, requires that the cultured animals have a health status and genetic characteristics suited to thriving in these environments. Shrimp breeding lines developed for high investment super-intensive grow-out systems will continue to, and increasingly so, focus on high-density survival and growth in order to achieve commercial viability. However, breeding lines with enhanced pathogen tolerance and resilience traits will likely also have an important role for farmers operating systems less amenable to biosecurity, as an insurance measure to mitigate pathogen incursions and disease. Operating in parallel to breeding, the risks of pathogens outbreaks and disease can be reduced by proper feed management and inclusions of feed additives and functional ingredients in formulations. Optimized nutrition (selection of feeds based on their composition and functionality) and feeding practices (feeding ration and frequency) suited to the farms’ production system, all operating within an appropriate business planning framework, will best position super-intensive farming operations for success. Moreover, more economic analysis of commercial shrimp systems is needed to provide ongoing guidance into how best to develop sustainable and optimal management practices for super-intensive farming.

Finally, the super-intensive farming sector needs to be proactive in improving its credentials in relation to the different dimensions of sustainability and social license. Development of super-intensive farming in ways that align to enhanced environmental sustainability, and which consider growing consumer concerns of animal welfare, product quality, and food safety, will be important to avoid excessive critique of the industry; to allow ongoing access of super-intensively produced products to many global markets; and to ensure the industry takes a development path that is responsible and looking to the future.

## Figures and Tables

**Figure 1 animals-12-00236-f001:**
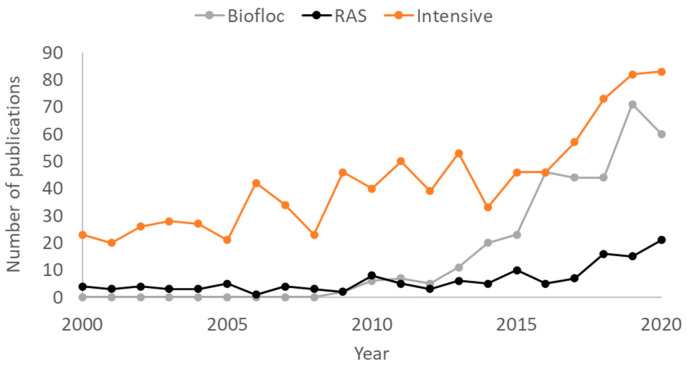
Bibliometric analyses of the evolution of scientific studies on penaeid intensification (using the combined search terms “Shrimp + biofloc”, “Shrimp + RAS” and “Shrimp + intensive”, in Scopus website).

**Figure 2 animals-12-00236-f002:**
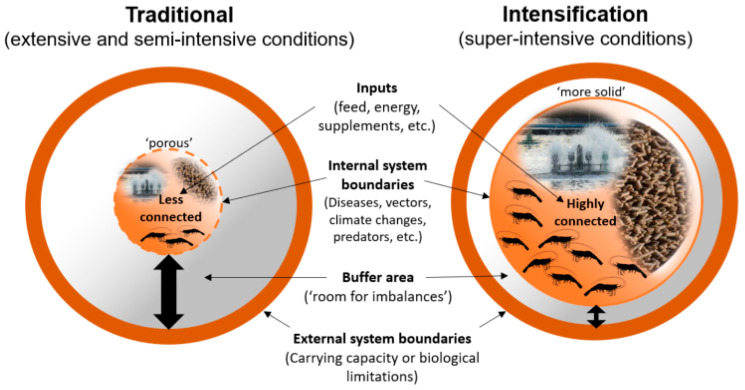
Comparison between traditional (extensive or semi-intensive) and super-intensive shrimp systems. The higher the inputs and degree of intensification, the less room for error.

**Figure 3 animals-12-00236-f003:**
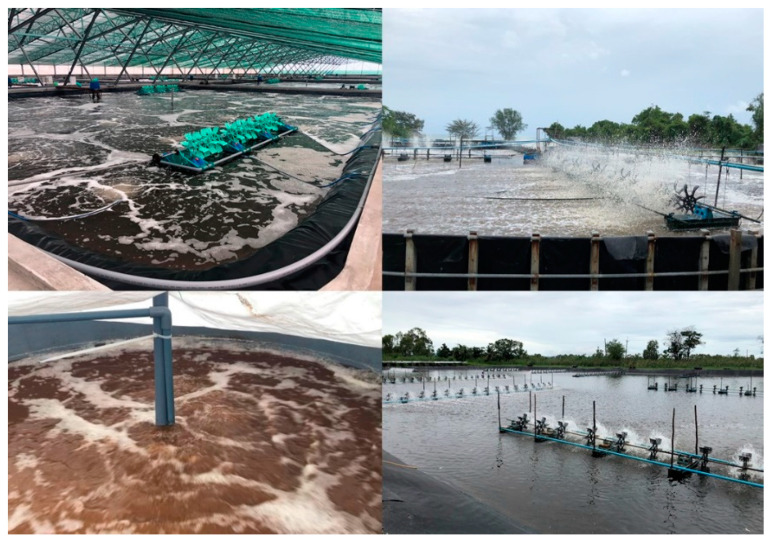
Examples of commercial microbial-based systems: (**upper left**) super-intensive indoor operations using chemoautotrophic-based BFT in Vietnam; (**upper right**) semi-biofloc (heterotrophic-based) in Thailand; (**lower left**) bioreactor being used in rice bran-based synbiotics in Vietnam; and (**lower right**) Aquamimicry integrated with tilapia in Thailand.

**Figure 4 animals-12-00236-f004:**
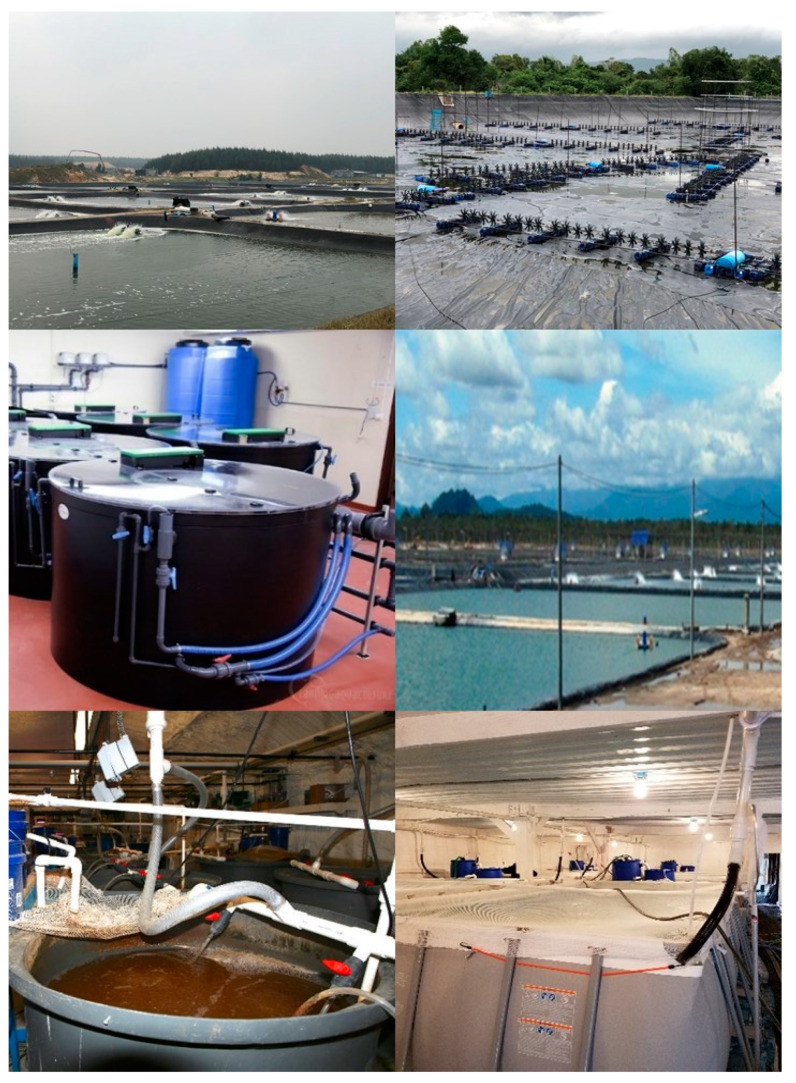
Super-intensive commercial shrimp production using flow-throw water in China and Thailand [28] (**upper**), pilot-scale indoor RAS used for *L. vannamei* nursery in Europe ([Source: C. Espinal and [82]) and outdoor RAS in a commercial shrimp farm, Malaysia [80] (**middle**); hybrid BioRAS at Lab-scale and commercial indoor farm, U.S. (**bottom**) (courtesy: A. Ray).

**Figure 5 animals-12-00236-f005:**
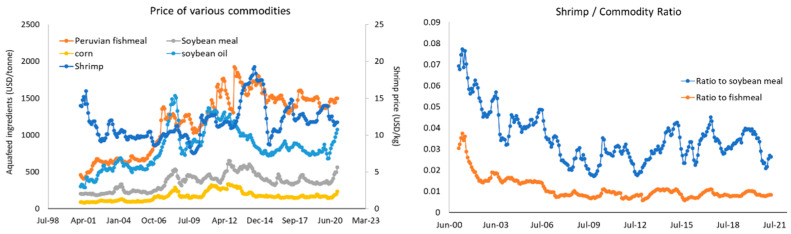
Price of various raw feed ingredients and shrimp (shell-on, headless, 26 to 30 counts per pound) as a food commodity (**left**); and shrimp to commodity ratio for two key raw ingredients, fishmeal, and soybean meal (**right**). Source: adapted from IndexMundi data [125].

**Figure 6 animals-12-00236-f006:**
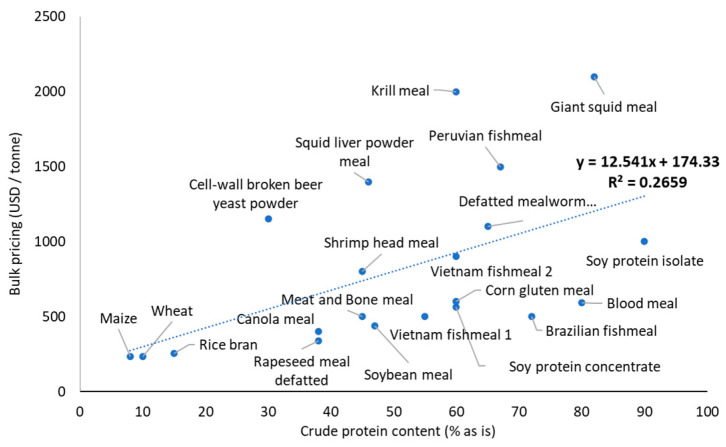
Price estimates for various raw feed and food-grade ingredients commercially available against their crude protein content estimates. Note ingredients were chosen from companies able to sustain 3000 to 120,000 tonnes per year production from Alibaba.com and their respective bulk FOB pricing.

**Table 1 animals-12-00236-t001:** Summary of biofloc technology and similar microbial-based approaches used in fully lined ponds or tanks, showing the main characteristics of the systems and references. The information presented below is a general guide only. Specific characteristics may change according to different culture conditions and management strategies.

System	Main Characteristics	Reference
1. Biofloc technology (heterotrophic-based, also known as ‘pure BFT’)	Several scientific studies available High reliance on heterotrophic bacteria to control toxic N-compounds Application of a mature (biofloc-based) inoculum is often observed to speed up the microbial maturation process High C:N ratio (normally 15–20:1) and routine external carbon inputs (normally during the entire cycle) Levels of suspended solids (e.g., settling solids) normally varying from 5 up to more than 10 mL/L Clarifiers, protein skimmers, and/or settling chambers to control suspended solids are often used Low-intermediate water exchange rates to remove the sludge and solids Drop in pH and alkalinity is often observed Routine application of water probiotics, carbonate and bicarbonate sources and other minerals Biofloc particles with low lipid content	[3,4,9,73,74,75,76]
2. Biofloc technology (chemoautotrophic based)	Little scientific information available High reliance on chemoautotrophic (nitrifying) bacteria to control N-compounds Application of a mature chemoautotrophic-based inoculum is often observed to speed up the microbial maturation process Intermediate C:N ratio (~10:1) with low external carbon inputs (normally up to the first 30–50 days) or even none Low levels of suspended solids (e.g., settling solids, normally up to 5 mL/L) Clarifiers, protein skimmers, and/or settling chambers to control suspended solids are often used Drop in pH and alkalinity is often observed Routine application of water probiotics and intensive control of alkalinity levels with routine application of carbonate and bicarbonate and other mineral sources. Low water exchange rates to remove the sludge and solids Biofloc particles with low lipid content	[4,17,74,77]
3. Semi-biofloc (photoautotrophic-based, also known as ‘green-biofloc’)	Little scientific information available High reliance on microalgae to control N-compounds, resulting in low capacity of organic matter digestion by heterotrophic (degrading) bacteria Adoption of a mature inoculum is quite unusual Little control of the C:N ratio (normally < 10:1) with low or no external carbon inputs Low levels of suspended solids (e.g., settling solids normally up to 5 mL/L, but higher levels can be observed with algae blooms) Clarifiers, protein skimmers, and/or settling chambers to control suspended solids are quite unusual Partial/none mesh cover in ponds with some degree of pH fluctuation Routine application of water probiotics. However, less need for carbonate and bicarbonate sources to control alkalinity Higher water exchange rates to remove the sludge (mainly dead algae cells) and suspended solids Biofloc particles with high lipid content	[3,37,78]
4. Semi-biofloc (mixed trophic conditions)	Little scientific information available A mix between algae and bacteria is promoted to control N-compounds, resulting in intermediate capacity of organic matter digestion by heterotrophic (degrading) bacteria Adoption of a mature inoculum is quite unusual Intermediate C:N ratio (10–15:1) with external carbon inputs, especially in the first days of production cycle Low levels of suspended solids (e.g., settling solids normally up to 5 mL/L) Clarifiers, protein skimmers, and/or settling chambers to control suspended solids are quite unusual Ponds are fully or partially mesh covered Large amount of water probiotics is routinely applied into ponds, helping to control algae bloom and pH fluctuations. Less need of carbonate and bicarbonate sources to control alkalinity Intermediate to high water exchange rates to remove the sludge and suspended solids Biofloc particles with intermediate lipid content	[3,66,67]
5. Aquamimicry (without fish)	Little scientific information available A mix between algae and bacteria is promoted to control N-compounds, resulting in intermediate capacity of organic matter digestion by heterotrophic (degrading) bacteria Routine external fermented carbon inputs generate intermediate C:N ratio (10–15:1) Routine application of water probiotics, and carbonate and bicarbonate sources to control alkalinity Normally low levels of suspended solids (e.g., settling solids normally up to 5 mL/L, but higher levels can be observed with algae blooms) Clarifiers, protein skimmers, and/or settling chambers to control suspended solids are quite unusual Intermediate to high water exchange rates to remove the sludge and suspended solids Zooplankton blooms (e.g., insect larvae, copepods, rotifers, etc.) are expected	[70]
6. Aquamimicry (integrated with fish)	Little scientific information available Water continuously circulates from shrimp ponds, to fish ponds and water treatment ponds with shrimp sludge been directed into fish ponds Other characteristics are the same as per ‘Aquamimicry without fish’.	[28]
7. AquaScience^®^ (integrated with tilapia)	Little scientific information is available A mix between algae and bacteria is promoted to control N-compounds. Shrimp sludge is drained (shrimp toilet) to fish and nitrification pounds. Decantation, heterotrophic bacteria, and microalgae are used to treat the water. After treatment, the water returns to shrimp pounds. Water exchange is minimal and is reused for consecutive production cycles. Relatively small and lined pounds are used for shrimp production (0.4 ha).	[71]
8. Synbiotics	Little scientific information available A mix between algae and bacteria is promoted to control N-compounds. High capacity of organic matter digestion by heterotrophic (degrading) bacteria Routine addition of external fermented (aerobic, anaerobic or both) carbon inputs, with or without the addition of exogenous enzymes, normally representing an intermediate C:N ratio (10–15:1) Low levels of suspended solids are expected (e.g., settling solids normally up to 5 mL/L) Clarifiers, protein skimmers and/or settling chambers to control suspended solids are unusual Intermediate water exchange rates to remove the sludge and suspended solids Routine application of water probiotics, carbonate, and bicarbonate sources to control alkalinity (pH is normally stable) High bacteria loads, intense zooplankton bloom might not occur	[68,69]

**Table 2 animals-12-00236-t002:** Summary of flow-through, RAS, and hybrid systems showing the main characteristics of the systems and references. The information presented is a general guide only and may change according to regional conditions and culture strategies.

System	Main Characteristics	Reference
1. Flow-through	Little scientific information available Water exchange varying from 10 to more than 100%/day is promoted to control N-compounds Relatively low natural productivity (e.g., bacteria, phytoplankton, and zooplankton) Low inputs (e.g., bioremediators, carbon sources, etc.) compared to microbial-based systems pH fluctuations and low levels of solids	[28,80]
2. RAS (recirculating aquaculture systems)	Little scientific information available Incorporates conventional RAS equipment and filtering devices (normally in indoor conditions, ‘boutique type’ farms, improving carrying capacity, e.g., >5 kg m^−3^) Relatively low natural productivity (e.g., bacteria, phytoplankton, and zooplankton) Routine application of carbonate and bicarbonate sources (use of water probiotics are quite unusual) Low water exchange rates	[80,81,82,85]
3. Green-water RAS (photoautotrophic-based)	No scientific information available Incorporates RAS equipment and filtering devices (indoor conditions) Phytoplankton and chemoautotrophic bacteria dominance over heterotrophic bacteria (less light control). Low capacity of organic matter digestion by heterotrophic (degrading) bacteria Little control of the C:N ratio (normally < 10:1) with low or no external carbon inputs Higher water circulation rates to remove the sludge (mainly dead algae cells) and suspended solids (e.g., settling solids normally up to 5 mL/L, but higher levels can be observed with algae blooms) Routine application of carbonate and bicarbonate sources, and usual application of water probiotics Low water exchange rates	[86]
4. BioRAS (heterotrophic based)	Little scientific information available Incorporates RAS equipment and filtering devices (indoor conditions) Heterotrophic and chemoautotrophic bacteria dominance over phytoplankton (higher light control, pH stable) Intermediate C:N ratio (~10:1) with low external carbon inputs (normally up to the first 30–50 days) Intermediate capacity of organic matter digestion by heterotrophic (degrading) bacteria Higher water circulation rates to remove the sludge and suspended solids (e.g., settling solids normally up to 5 mL/L) Routine application of bioremediators, carbonate and bicarbonate sources Low water exchange rates	[83,85,87]

**Table 4 animals-12-00236-t004:** Summary of recent studies evaluating different aspects of *L. vannamei*-based integrated rearing systems.

Integrated Species	Shrimp Production System	Evaluated Aspect	Main Findings	Reference
**Aquaponics**				
*Sarcocornia ambigua*	BFT	Aquaponics vs. shrimp monoculture	N use was 25% more efficient, 2 kg of plants produced for each kg of shrimp	[107]
*Ocimum basilicum*	RAS	Water source and aquaponic system	Low-salinity groundwater resulted in greater shrimp and basil yields	[108]
*S. ambigua*	BFT	Different salinities	Optimal salinity between 16 and 24 g L^−1^	[114]
**Shrimp and Macroalgae**				
*Ulva lactuca*	Recirculation system	Integration vs. shrimp monoculture	Integrated system maintained adequate water quality, improved growth for shrimp fed seaweed	[111]
*U. prolifera*	Minimum water exchange	Water exchange rate and algae density	10% water exchange and 800 mg L^−1^ of stocked algae improved shrimp growth and survival	[115]
*U. fasciata* and *U. ohnoi*	BFT	Algae species and density	Best performance for *U. ohnoi* under 2 g L^−1^	[110]
**Shrimp and fish or shellfish**				
*O. niloticus*	BFT	Fish stocking densities	Recovery of N and P and overall yield increased linearly	[116]
*M. curema*	BFT	Integration vs. shrimp monoculture	Increases in overall yield and P recovery	[117]
*M. liza*	BFT	Integration vs. shrimp monoculture	Lower TSS concentrations in integrated systems	[113]
*M. curema*	BFT	Fish stocking densities	Integration of mullet and shrimp increased biofloc system yield by 11.9%	[118]
*O. niloticus*	BFT	Heterotrophic and mature BFT systems	Higher fish and overall yields in heterotrophic BFT	[119]
*Crassostrea gigas*	Water exchange-based	Integration vs. shrimp monoculture	Improved shrimp growth performance and water quality by oyster presence	[120]
*O. niloticus*	BFT	Fish stocking densities	Increasing stocking densities affected fish physiology	[121]
**Multitrophic**				
*O. niloticus and S. ambigua*	BFT	Three-species integration vs. shrimp+fish	Total yield increased by 21.5%, reduction in water nitrate concentration	[112]
*M. liza and U. fasciata*	BFT	Three-species integration vs. shrimp+fish	Yield increase of 12.2%, improved N and P recovery, improved sea lettuce biochemical composition	[122]

BFT: Biofloc technology; RAS: recirculating aquaculture system; TSS: total suspended solids.

**Table 5 animals-12-00236-t005:** Comparison of recommended minimum nutrient requirements in diets for *L. vannamei* in different production systems.

Nutrient Requirements (%)	*L. vannamei*		
	RAS	Semi-Intensive	Intensive
Crude protein	38–44	33–42	40–46
Crude lipid	9–11	7	8
Dig. energy (kJ/kg)	15,820–16,292	14,033–15,380	15,079–15,874
Amino acids (%)			
Arg	2.56–2.94	2.58–2.92	2.69–2.99
His	0.73–0.83	0.73–0.82	0.77–0.84
Ile	1.51–1.71	1.52–1.70	1.59–1.73
Leu	2.52–2.99	2.53–2.98	2.64–3.06
Lys	2.76–3.18	2.72–3.14	2.83–3.22
Met	0.97–1.11	0.98–1.11	1.01–1.13
Phe	1.74–1.97	1.76–1.96	1.83–2.00
Thr	1.31–1.56	1.31–1.54	1.37–1.58
Trp	0.34–0.39	0.34–0.39	0.36–0.39
Val	1.7–2.01	1.72–2.00	1.79–2.04
Fatty acids (%)			
Sum n-3	0.89	0.83	0.87
Sum n-6	0.6	0.6	0.6
EPA + DHA	0.71–1.01	0.67–0.94	0.69–0.98
Cholesterol	667–834	521–727	540–752
Phospholipids	1.1–1.5	1–1.4	1.1–1.4

Adapted from International Aquaculture Feed Formulation Database (IAFFD). Values represent minimum requirements across all life stages of *L. vannamei* (<1 g pre-start to >12 g finisher), available from [132]. Values are estimated through advanced nutritional modeling efforts based on the effective compiling, integrating, statistical analysis, and interpreting available research-based and production-specific data.

**Table 6 animals-12-00236-t006:** Apparent protein digestibility (APD) of ingredients for *L. vannamei*.

Ingredient	APD (%)	Reference
Fishmeal	83.7–91.6	[150,151,153,164]
Krill meal	80.5	[151]
Meat and bone meal	73.9–82.2	[150,153]
Hydrolyzed feather meal	63.9	[151]
Poultry meal	75.0–78.7	[150,151,153]
Soybean meal	89.0–96.9	[150,151,153,154]
Soy protein isolate	93.7–96.2	[151,154]
Canola meal	78.3	[150]
Wheat (gross energy digestibility)	87.0	[155]

**Table 7 animals-12-00236-t007:** Examples of commercial feed additives available for the shrimp industry.

Supplier	Adisseo	Biomin	Alltech	Lallemand	Evonik	DSM	Cargill	BASF	DuPont (Danisco Animal Nutrition)	ADM	Kemin
Prebiotic, immunostimulants and immunomodulators	Nutri^®^-Farm Stim S	Levabon^®^ Aquagrow E	Bio-MOS^®^, Actigen^®^	Agrimos, Yang, M-glucan, Fibosel				BalanGut^®^ LS		CitriStim^®^	Aquastem^TM^
Probiotic	Nutri^®^-Farm P/PW/L/FE	AquaStar^®^	Acid-Pak 4 way Lacto-Sacc	Bactocell	Ecobiol^®^, Fecinor^®^, Gutcare^®^						
Phytobiotic	Sanacore^®^ GM		Natustat							Xtract (Pancosma)	
Amino acids	Rhodimet^®^ For Aqua				MetAMINO^®^, AQUAVI^®^ Met-Met, Biolys^®^, ThreAMINO^®^, TrypAMINO^®^				Betafin^®^	Proplex^®^, L-lysine, L-threonine	
Nucleotides			Nupro^®^	Laltide^®^		Rovimax, Rovimax NX Plus					
Enzymes and digestion enhancers	Aquagest^®^ S, Aqualyso^®^, Lipogest^®^	Digestarom^®^	Aquate, Allzyme SSF^®^ Allzyme VegPro			Phytase, xylanase and protease WX RONOZYME^®^ WX, HiPhos RONOZYME^®^ ProAct, PRoAct 360TM			Xylanase, phytase		Aquatria^TM^
Vitamins	Aquavit^®^ C stable					OVN, ROVIMIX^®^ STAY-C^®^35		Lutavit^®^ A, Lutavit^®^ B2, Lutavit^®^ E, Vitamin A palmitate, propionate, acetate			Vibrell^TM^ C
Minerals	Selisseo^®^		Bioplex^®^, Sel-plex^®^					Copper-glycinate, Iron-glycinate, manganese-glycinate, Zinc-glycinate		B-traxim (Pancosma)	
Acidifiers (organic acids)	Bacti-Nil^®^	Biotronic^®^	Acid-Balance AcidPak 4 Way					Amasil^®^, Lupro-Cid^®^, Lupro-Grain^®^, Lupro-Mix^®^NA,		DaaFIT® (Pancosma)	
Antioxidants	Oxy-Nil^®^ Aqua Zero,		Banox Antiox-Av Antiox-RC Nature-Ban Vitalix	Alkosel 2000, Melofeed		Carophyll Pink Rovimix E Stay C	Proviox^TM^	Lucanthin^®^ Pink			Oxivia^TM^ C
Attractants	Aquabite^®^										
Binders	Nutribind^®^		AllBind								

## Data Availability

Not applicable.

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
