# Peer review of "Intensification of Penaeid Shrimp Culture: An Applied Review of Advances in Production Systems, Nutrition and Breeding"

_animals, 2022, doi:10.3390/ani12030236_

Round 1
Reviewer 1 Report
Very interesting article and written with great care.
The scientific names still need to be reviewed, they sometimes appear without italics and the first time they are used they should be in full ex, line 56 L. vannamei should be in full and italics.
In the keywords, I suggest that the scientific name should also be complete, the full scientific name, as well as in the figure and table legends.
Author Response
Dear reviewers. Thank you for your valuable contributions that certainly improved the manuscript quality. The authors addressed all comments and suggestions (highlighted below and main text). Specific responses can also be found below. Kind regards.
Reviewer #1:
Very interesting article and written with great care. R: Authors appreciate your comment. Thank you.
The scientific names still need to be reviewed, they sometimes appear without italics and the first time they are used they should be in full ex, line 56 L. vannamei should be in full and italics.
In the keywords, I suggest that the scientific name should also be complete, the full scientific name, as well as in the figure and table legends. R: Thank you. We reviewed and adjusted them all across the entire manuscript (including references).

Reviewer 2 Report
Review on Intensification of Penaeid shrimp culture: an applied review of advances in production systems, nutrition and breeding (animals-1509923) by M.G.C. Emerenciano and co-authors.
The authors made a comprehensive review on the intensification of Penaeid shrimp culture, focusing on important aspects such as production systems, nutrition, breeding, health and welfare, and sustainability. The review is in general well-organised and well-written and is an excellent overview of the actual status of this industry.
There are some minor comments on specific aspects of the review and a revision most Tables is advised.
- L. vannamei and P. monodon should be in italics along the whole manuscript. Other scientific names are also not in italics.
- Line 99: Is this correct: m3 or hectare, or should it be m2?
- Line 101: what do the authors mean by supplements? Please provide some examples and/or specify which type of supplements are you referring to.
- Table 1 would benefit from some organization. For instance, in 1. Biofloc technology: “drop in pH and alkalinity is often observed with routine application of carbonate and bicarbonate sources (…)”; but then, in 2. Biofloc technology: “intensive control of alkalinity levels with routine application of carbonate and bicarbonate (…) Drop in pH is often observed”. Standardisation of sentences will improve reading. Furthermore, if you maintain the same organisation each production system, it would be easier to identify differences among them. For instance: water exchange rates, levels of suspended solids, biofloc characteristics, etc (systems 3, 4 and 8 are somehow better organised).
- Line 275: figure is not mentioned in the text.
- Table 2: System 3 has no scientific literature available, so what was the source of information used?
- Line 370: specify which are the other two materials.
- Table 3 would benefit from a better organisation. For instance, organised according to the evaluated aspects.
- Table 4: please revise in page 14 the main findings for M. curema. Something seems to be missing.
- Line 431: replace good by balanced.
- Line 468: please rephrase: 32.9% is not exactly little dietary protein.
- Lines 478-479: revise this sentence and/or add more information, since this practice will lead to increased environmental impact due to higher nitrogen losses resulting from protein catabolism.
- Table 5: change kcal to kJ, replace Isl by Ile and Try by Tyr.
- Line 502: not clear why Table 5 is mentioned.
- Line 682: delete rates.
- Line 794: delete sequencing after HTS.
Author Response
Dear reviewers. Thank you for your valuable contributions that certainly improved the manuscript quality. The authors addressed all comments and suggestions (highlighted below and main text). Specific responses can also be found below. Kind regards.
Reviewer #2:
Review on Intensification of Penaeid shrimp culture: an applied review of advances in production systems, nutrition and breeding (animals-1509923) by M.G.C. Emerenciano and co-authors.
The authors made a comprehensive review on the intensification of Penaeid shrimp culture, focusing on important aspects such as production systems, nutrition, breeding, health and welfare, and sustainability. The review is in general well-organised and well-written and is an excellent overview of the actual status of this industry. R: Authors appreciate your comment. Thank you.
There are some minor comments on specific aspects of the review and a revision most Tables is advised.
- L. vannamei and P. monodon should be in italics along the whole manuscript. Other scientific names are also not in italics. R: We reviewed and adjusted them all across the entire manuscript. Thank you.
- Line 99: Is this correct: m3 or hectare, or should it be m2? R: Thank you. We changed it accordingly.
- Line 101: what do the authors mean by supplements? Please provide some examples and/or specify which type of supplements are you referring to. R: Thank you. We provided examples.
- Table 1 would benefit from some organization. For instance, in 1. Biofloc technology: “drop in pH and alkalinity is often observed with routine application of carbonate and bicarbonate sources (…)”; but then, in 2. Biofloc technology: “intensive control of alkalinity levels with routine application of carbonate and bicarbonate (…) Drop in pH is often observed”. Standardisation of sentences will improve reading. Furthermore, if you maintain the same organisation each production system, it would be easier to identify differences among them. For instance: water exchange rates, levels of suspended solids, biofloc characteristics, etc (systems 3, 4 and 8 are somehow better organised).
R: Valuable suggestion. The authors reviewed and reorganised the Tables. Thank you.
- Line 275: figure is not mentioned in the text. R: Authors adjusted it. Thank you.
- Table 2: System 3 has no scientific literature available, so what was the source of information used? R: Personal communication (G.L.Mello). The information was included in the text. Thank you.
- Line 370: specify which are the other two materials. R: Information was included
- Table 3 would benefit from a better organisation. For instance, organised according to the evaluated aspects. R: Table was adjusted and now is better organised. Thank you.
- Table 4: please revise in page 14 the main findings for M. curema. Something seems to be missing. R: Information was adjusted/corrected. Thank you.
- Line 431: replace good by balanced. R: Information was adjusted/corrected. Thank you.
- Line 468: please rephrase: 32.9% is not exactly little dietary protein. R: Information was adjusted/corrected. Thank you.
- Lines 478-479: revise this sentence and/or add more information, since this practice will lead to increased environmental impact due to higher nitrogen losses resulting from protein catabolism. R: Thank you. The information was added.
- Table 5: change kcal to kJ, replace Isl by Ile and Try by Tyr. R: The information (Table 5) was adjusted accordingly. Thank you.
- Line 502: not clear why Table 5 is mentioned. R: Thank you. The text was restructured and additional information related to Table 5 added.
- Line 682: delete rates. R: The text was adjusted accordingly. Thank you.
- Line 794: delete sequencing after HTS. R: Deleted.

Reviewer 3 Report
The manuscript reviewed production systems of shrimp of which production and demand are increasing all over the world. Especially, focusing on the super-intensive systems, it summarizes the outline of the system, nutrition, breeding, social problem while comparing it with the traditional system and semi-intensive system. This manuscript referred many papers, and I learned a lot. However, the manuscript has some points to must confirm and to revise.
Major point is that it needs to clarify whether the super-intensive systems is for L. vannamei. If some referred paper targets other shrimp species, please show it in table 1 and 2. And, is this system sand-free? Moreover, because the title contains “Penaeid”, advantages and problems of land production system of other shrimps (e.g. kuruma shrimp required sand. It is expected that the system with sand has some specific problems.) should be contained. In section 3, is there paper and program which inspected insect (e.g. meal worm) as alternative protein?
Minor point
・What kind of effect is a polyester substrate in line 369. It is better to show a little more information.
・Please separate by line or leave a space between each system in table 1, 2, 3, 4, and 7.
・Please confirm abbreviation in line 464, this is the first time in manuscript.
・Please refer figure 3 and 6 in text.
・In figure 5 and 6, letters in figure are small.
Author Response
Dear reviewers. Thank you for your valuable contributions that certainly improved the manuscript quality. The authors addressed all comments and suggestions (highlighted below and main text). Specific responses can also be found below. Kind regards.
Reviewer #3:
The manuscript reviewed production systems of shrimp of which production and demand are increasing all over the world. Especially, focusing on the super-intensive systems, it summarizes the outline of the system, nutrition, breeding, social problem while comparing it with the traditional system and semi-intensive system. This manuscript referred many papers, and I learned a lot. R: Authors appreciate your comment. Thank you. However, the manuscript has some points to must confirm and to revise.
Major point is that it needs to clarify whether the super-intensive systems is for L. vannamei. If some referred paper targets other shrimp species, please show it in table 1 and 2. And, is this system sand-free? Moreover, because the title contains “Penaeid”, advantages and problems of land production system of other shrimps (e.g. kuruma shrimp required sand. It is expected that the system with sand has some specific problems.) should be contained. R: Thank for your comment. The article is focused on L. vannamei super-intensive production, under conditions of high stocking density (>150/m2) and lined ponds/tank-based. This information was better evidenced and highlighted in the ‘objectives’ (L140).
In section 3, is there paper and program which inspected insect (e.g. meal worm) as alternative protein? R: Thank for your comment. Likely yes, but authors understand that such topic is not exactly the focus of such section. Thank you.
Minor point
・What kind of effect is a polyester substrate in line 369. It is better to show a little more information. R: More information was added. The NeedLona provided higher survival and lower concentrations of TSS (mentioned in the text). Also more additional information was added related to other two substrate materials.
・Please separate by line or leave a space between each system in table 1, 2, 3, 4, and 7. R: Thank you. Tables were restructured accordingly.
・Please confirm abbreviation in line 464, this is the first time in manuscript. R: Abbreviations were reviewed and adjusted accordingly. Thank you.
・Please refer figure 3 and 6 in text. R: Figures were referred. Thank you.
・In figure 5 and 6, letters in figure are small. R: Letters were adjusted accordingly. Thank you.
